# Histoplasmosis Diagnosed in Europe and Israel: A Case Report and Systematic Review of the Literature from 2005 to 2020

**DOI:** 10.3390/jof7060481

**Published:** 2021-06-14

**Authors:** Spinello Antinori, Andrea Giacomelli, Mario Corbellino, Alessandro Torre, Marco Schiuma, Giacomo Casalini, Carlo Parravicini, Laura Milazzo, Cristina Gervasoni, Anna Lisa Ridolfo

**Affiliations:** 1Luigi Sacco Department of Biomedical and Clinical Sciences, Università di Milano, 20157 Milan, Italy; andrea.giacomelli@unimi.it (A.G.); marco.schiuma@unimi.it (M.S.); giacomo.casalini@unimi.it (G.C.); 2III Division of Infectious Diseases, ASST Fatebenefratelli Sacco, 20157 Milan, Italy; mario.corbellino@asst-fbf-sacco.it (M.C.); alessandro.torre@asst-fbf-sacco.it (A.T.); laura.milazzo@asst-fbf-sacco.it (L.M.); cristina.gervasoni@asst-fbf-sacco.it (C.G.); annalisa.ridolfo@asst-fbf-sacco.it (A.L.R.); 3Pathology Unit, ASST Fatebenefratelli Sacco, 20157 Milan, Italy; carlo288@gmail.com

**Keywords:** histoplasmosis, *Histoplasma capsulatum*, *Histoplasma duboisii*, HIV, immunocompromise, travellers, progressive disseminated histoplasmosis, diagnosis

## Abstract

Human histoplasmosis is a mycosis caused by two distinct varieties of a dimorphic fungus: *Histoplasma capsulatum* var. *capsulatum* and *H. capsulatum* var. *duboisii*. In Europe, it is usually imported by migrants and travellers, although there have been some autochthonous cases, especially in Italy; however, most European physicians are unfamiliar with its clinical and pathological picture, particularly among immunocompromised patients without HIV infection. This systematic review of all the cases of histoplasmosis reported in Europe and Israel between 2005 and 2020 identified 728 cases diagnosed in 17 European countries and Israel described in 133 articles. The vast majority were imported (mainly from Central and South America), but there were also seven autochthonous cases (six in Europe and one in Israel). The patients were prevalently males (60.4%), and their ages ranged from 2 to 86 years. The time between leaving an endemic region and the diagnosis of histoplasmosis varied from a few weeks to more than 40 years. Progressive disseminated histoplasmosis was the most frequent clinical picture among people living with HIV infection (89.5%) or a different immunocompromising condition (57.1%), but it was also recorded in 6.2% of immunocompetent patients. Twenty-eight cases were caused by *Histoplasma duboisii*. Immunocompromised patients without HIV infection had the worst outcomes, with a mortality rate of 32%.

## 1. Introduction

Human histoplasmosis is a disease that is traditionally considered to be caused by two varieties of the fungus *Histoplasma capsulatum* var. *capsulatum* (*Hcc*) and *H. capsulatum* var. *duboisii* (*Hcd*, also known as African histoplasmosis) [1,2]. Although the number of species belonging to the *Histoplasma* genus is unknown, the findings of genome-wide population genetic and phylogenetic analyses suggest the existence of four species that are endemic in the Americas: *Histoplasma capsulatum sensu stricto*, *H. mississippiense* sp. nov (formerly known as NAm1), *H. ohiense* sp. nov (formerly known as NAm2), and *H. suramericanum* sp. nov (formerly known as LAm A) [3]. A fifth lineage, the African clade (*H. capsulatum* var. *duboisii*), is probably a separate species, but confirmation of its status will require further analyses of more samples [3].

The disease is highly endemic in the USA and Latin America [4] but is probably more widespread than previously thought [5,6]; a survey by the European Confederation of Medical Mycology Working Group has described the cases diagnosed in Europe between January 1995 and December 1999 [7], and Antinori et al. have reviewed cases observed in patients living with HIV infection in Europe between 1984 and 2004 [8]. Other publications include a systematic review of cases of acute histoplasmosis in immunocompetent travellers worldwide [9] and a review of the worldwide cases of African histoplasmosis diagnosed between 1993 and 2019 [10].

The aim of this systematic review is to describe the imported and autochthonous cases of histoplasmosis diagnosed in Europe and Israel between 2005 and 2020 as well as a new case of progressive disseminated histoplasmosis in a Brazilian trans-sexual male living with HIV/AIDS.

## 2. Materials and Methods

This systematic review was carried out in accordance with the Preferred Reporting Items for Systematic Reviews and Meta-analyses (PRISMA) statement [11].

### 2.1. Search Strategy

The PubMed database was searched for articles published between January 2005 and December 2020 using combinations of key words: histoplasmosis (AND) Europe; *Histoplasma capsulatum* (AND) Europe; *Histoplasma duboisii* (AND) Europe; histoplasmosis, disseminated (AND) Europe; histoplasmosis, pulmonary (AND) Europe; histoplasmosis, extra-pulmonary (AND) Europe; histoplasmosis, autochthonous (AND) Europe; histoplasmosis, imported (AND) Europe; histoplasmosis, HIV (AND) Europe; histoplasmosis, immunocompetent (AND) Europe; histoplasmosis, immunocompromised (AND) Europe; histoplasmosis, travellers (AND) Europe; histoplasmosis (AND) Albania; histoplasmosis (AND) Andorra; histoplasmosis (AND) Armenia; histoplasmosis (AND) Austria; histoplasmosis (AND) Azerbaijan; histoplasmosis (AND) Belarus; histoplasmosis (AND) Belgium; histoplasmosis (AND) Bosnia (OR) Herzegovina; histoplasmosis (AND) Bulgaria; histoplasmosis (AND) Croatia; histoplasmosis (AND) Cyprus; histoplasmosis (AND) Czech; histoplasmosis (AND) Denmark; histoplasmosis (AND) Estonia; histoplasmosis (AND) Finland; histoplasmosis (AND) France; histoplasmosis (AND) Georgia; histoplasmosis (AND) Germany; histoplasmosis (AND) Greece; histoplasmosis (AND) Hungary; histoplasmosis (AND) Iceland; histoplasmosis (AND) Ireland; histoplasmosis (AND) Israel; histoplasmosis (AND) Italy; histoplasmosis (AND) Kazakhstan; histoplasmosis (AND) Kosovo; histoplasmosis (AND) Latvia; histoplasmosis (AND) Lichtenstein; histoplasmosis (AND) Lithuania; histoplasmosis (AND) Luxembourg; histoplasmosis (AND) Macedonia; histoplasmosis (AND) Malta; histoplasmosis (AND) Moldovia; histoplasmosis (AND) Montenegro; histoplasmosis (AND) Netherlands; histoplasmosis (AND) Norway; histoplasmosis (AND) Poland; histoplasmosis (AND) Portugal; histoplasmosis (AND) Romania; histoplasmosis (AND) Russia; histoplasmosis (AND) Serbia; histoplasmosis (AND) Slovakia; histoplasmosis (AND) Slovenia; histoplasmosis (AND) Spain; histoplasmosis (AND) Sweden; histoplasmosis (AND) Switzerland; histoplasmosis (AND) Turkey; histoplasmosis (AND) Ukraine; and histoplasmosis (AND) United Kingdom.

### 2.2. Eligibility Criteria and Study Selection

The populations considered were children and adults of any age and origin diagnosed with histoplasmosis in geographical Europe and Israel. All publications were considered regardless of their type (individual case reports, case series, observational studies, and reviews providing information regarding age, gender, nationality, travel history, affected organs, latency period, diagnosis, antifungal therapy, and outcome) or language (excluding Turkish, Finnish, Romanian, and Russian). The cases included travel-related, possible autochthonous, and veterinary cases diagnosed in the same geographical areas.

The patients were divided into three categories based on their immune status: people living with HIV (PLWH); patients with other immunocompromising conditions (OIC: solid organ or stem cell transplant recipients; cancer patients; patients treated with immunosuppressive drugs including corticosteroids, biological immunosuppressants, and non-biological, non-corticosteroidal drugs); and immunocompetent subjects (HIV-negative patients not meeting any of the OIC criteria).

### 2.3. Data Collection and Evidence Summary

The following information was extracted from each article and entered into pilot-tested evidence tables: author, year, study design, language, country of diagnosis, country of exposure, latency period, number of cases, patients’ characteristics (age, gender, occupation, and affected organs), systemic antifungal therapy, and outcomes.

## 3. Results

### 3.1. Case Report

A 27 year old trans-sexual male born in Brazil was admitted to our infectious diseases ward in April 2012 with a 2-month history of intermittent high fever, productive cough, dysphagia, and diarrhoea. He had been concomitantly diagnosed with human immunodeficiency virus-1 (HIV-1) infection and disseminated tuberculosis in February 2006, after which he underwent a successful 12-month course of treatment for tuberculosis and antiretroviral therapy (ART: initially tenofovir/emtricitabine plus efavirenz followed by tenofovir/emtricitabine plus atazanavir/ritonavir), which led to a good recovery of CD4+ lymphocytes from 39/μL to 442/μL and controlled HIV-1 viremia (HIV RNA copy numbers went from 500,000 to <50/mL) until February 2011, when he was lost to follow-up.

A physical examination at the time of admission revealed a high fever (39.7 °C), cervical lymphadenopathy, hepatosplenomegaly, and oral candidiasis. His white blood cell count was 2530/μL (85% neutrophils, 8% lymphocytes), haemoglobin level 8.8 g/dL and platelet level 63,000/μL. He had a CD4+ cell count of 8/μL, an HIV-1 viral load of 30,780 copies/mL, and a normal lactate dehydrogenase level of 247 IU/L. He was initially treated with amoxicillin/clavulanate because of suspected pneumonia revealed by chest X-ray and started ART with cotrimoxazole prophylaxis. He refused a bone marrow biopsy and, after defervescence for 6 days, developed hypotension, a high fever (up to 40 °C), and dyspnoea. Whole-body computed tomography (CT) revealed generalised bilateral cervical lymphadenopathy (nodes 1–3 cm in diameter), retroperitoneal lymph nodes, hepatosplenomegaly, and bilateral interstitial/alveolar infiltrates in both lungs. Following unsuccessful treatment with helmet continuous positive airway pressure (CPAP), the patient was transferred to our intensive care unit (ICU) and underwent orotracheal intubation. As his clinical picture aroused the suspicion of multicentric Castleman disease accompanied by high-level of human herpes virus 8 (HHV-8) viremia (3 × 10^5^ copies/μL), he underwent a lymph node biopsy and started treatment with intravenous etoposide (150 mg). Histopathological examination of the lymph nodes showed a total replacement of follicular and paracortical areas by a histiocytic infiltrate and PAS/Grocott-positive fungal cells compatible with *Histoplasma* spp. (Figure 1).

Pan-fungal polymerase chain reaction (PCR) followed by sequencing of the amplified products confirmed the diagnosis of *Histoplasma capsulatum* infection. Serum galactomannan (GM) antigen (Platelia *Aspergillus* EIA, BioRad, Marnes la Coquette, France) was positive (OD index 2.73), and the patient was started on liposomal amphotericin B (L-AMB) at 3 mg/kg/day. After 12 days, he was extubated and returned to our ward. He completed the 2-week course of L-AMB and was started on itraconazole 200 mg every 12 h. Before discharge, he was administered etoposide again because of the reappearance of fever associated with a new increase in HHV-8 viremia levels, and ART was resumed with tenofovir/emtricitabine plus raltegravir and oral etoposide every week. In the following months, he returned to Brazil and was lost to follow-up.

The patient was re-hospitalised because of fever, abdominal pain, and cough in November 2012, when he said he had discontinued itraconazole for 3 months and ART for 1 week. Blood cultures were negative, and whole-body CT revealed enlarged latero-cervical, retroperitoneal, and inguinal lymph nodes. HHV-8 viremia was negative, HIV RNA was 6,707 copies/mL, and his CD4+ cell count was 81/μL. On the basis of GM antigen positivity (OD index 2.14), he was again treated with L-AMB, which led to the rapid disappearance of fever and abdominal pain and a reduction in his GM antigen level (OD index 1.13). He was discharged with a prescription of itraconazole 200 mg twice daily, cotrimoxazole prophylaxis, and ART.

He was hospitalised again in March 2013 because of a 2-week history of fever, chills, cough, and vomiting. A chest X-ray revealed an interstitial picture with mediastinal enlargement; GM antigen was negative (<0.5), and a Ziehl–Neelsen sputum test and PCR indicated *Mycobacterium tuberculosis.* Antitubercular treatment led to a good clinical response, and the patient was definitely lost to follow-up in June 2013.

### 3.2. Summary of the Literature

The literature search identified 133 articles about cases of human and veterinary histoplasmosis from 17 European countries and Israel (Figure 2).

### 3.3. Study Selection and Characteristics

As shown in Table 1, five studies involving 505 subjects (295/436 males, 67.7%) described cumulative experiences of histoplasmosis: two from Spain (one regarding imported endemic mycoses from 1997 to 2014 [12], the other relating to histoplasmosis in Spanish travellers to Latin America [13]), a retrospective study from metropolitan France reporting all the cases of histoplasmosis in PLWH diagnosed between 1985 and 2006 [14], and two about cases of acute histoplasmosis diagnosed in travellers in a single centre in Italy between 2005 and 2015 [15] and in Israel between 2000 and 2012 [16].

The case reports and small case series of histoplasmosis diagnosed between 2005 and 2020 [17,18,19,20,21,22,23,24,25,26,27,28,29,30,31,32,33,34,35,36,37,38,39,40,41,42,43,44,45,46,47,48,49,50,51,52,53,54,55,56,57,58,59,60,61,62,63,64,65,66,67,68,69,70,71,72,73,74,75,76,77,78,79,80,81,82,83,84,85,86,87,88,89,90,91,92,93,94,95,96,97,98,99,100,101,102,103,104,105,106,107,108,109,110,111,112,113,114,115,116,117,118,119,120,121,122,123,124,125,126,127,128,129,130,131,132,133,134,135,136] involved a total of 223 patients (145 males, 65%) with a median age of 38 years (range 2–86).

A majority of the patients included in all the articles were PLWH (365/728, 50.1%), 78 (10.7%) were patients with OIC, and 292 (40.1%) were immunocompetent.

### 3.4. People Living with HIV (PLWH)

Histoplasmosis was diagnosed in 113 PLWH described in 41 case reports and 13 case series [17,18,19,20,21,22,23,24,25,26,27,28,29,30,31,32,33,34,35,36,37,38,39,40,41,42,43,44,45,46,47,48,49,50,51,52,53,54,55,56,57,58,59,60,61,62,63,64,65,66,67,68,69]. Their median age was 37 years (range 2–63), and there was a prevalence of males (72, 63.7%). HIV infection was diagnosed concomitantly with histoplasmosis in 40 patients (52.6%), and 36 (47.4%) had been previously diagnosed as having HIV infection. No information concerning the time of HIV infection diagnosis was available in 37 cases. The latency period (i.e., the period between the last time spent in an endemic region and the diagnosis of histoplasmosis), which was available for 52 patients, ranged from 1 to 420 months (median 66 months, 5.5 years). Table 2 shows the patients’ characteristics. All the cases (except one autochthonous case observed in Italy) were considered imported: 54/109 infections (49.5%) were acquired in Central and South America, and 47 (43.1%) in sub-Saharan Africa. Forty-three percent of the Latin American cases were acquired in Ecuador (16/54, 29.6%) and Colombia (7/54, 13%), whereas Ghana (9/47, 19.1%) and Ivory Coast (7/47, 14.9%) contributed the highest proportions of the cases acquired in Africa. The European countries in which 80.7% of the cases were diagnosed were Spain (56, 49.1%), France (22, 19.3%) and Italy (14, 12.3%).

Progressive disseminated histoplasmosis (PDH) was diagnosed in 102 of the 114 cases (89.5%); the remaining 12 patients presented with colonic or small bowel involvement (6 cases, 5.3%) [17,18,21,31,34,68], lymph node involvement (2 cases, 1.8%) [24,28], and there was one case each of primary cutaneous [39], cerebral [64], pulmonary [36] or vaginal histoplasmosis [35]. Eight cases of immune reconstitution inflammatory syndrome (IRIS) were diagnosed following antiretroviral treatment [26,29,48,57,59], three patients had haemophagocytic syndrome [27,30,59], and one presented with posterior reversible encephalopathy syndrome (PRESS) [54]. Concomitant opportunistic infections and/or neoplasia were observed in 28.3% of the patients at the time of the diagnosis of histoplasmosis (Table 2) [18,19,22,23,25,28,30,32,33,34,37,39,41,44,45,49,50,51,53,56,58,59,65,69]: ten patients had candidiasis [19,22,25,41,44,45,51,56,58,59], five *Pneumocystis jirovecii* pneumonia [23,50,51,69], two disseminated cryptococcosis [18,52], and one pulmonary coccidioidomycosis [22]. Kaposi’s sarcoma was observed in three patients [19,22,34]. Anaemia was reported in 94.7% of the patients (37/38), leukopenia in 70.3% (26/37), and thrombocytopenia in 70.4% (19/27). Table 3 shows the signs and symptoms of histoplasmosis and the organ involvement demonstrated by means of cultures or histology.

The diagnostic work-up varied: microscopic visualisation of the fungus was the main method of identification (66 cases, 58.4%), associated with culture (32 cases), PCR (12 cases), or both (8 cases); a culture was available in 65 cases (57.5%). PCR was reported in 43 cases (38%) and was the only method used in three (2.6%). *Histoplasma* spp. was directly visualised on peripheral blood smears of eight patients [19,30,40,41,45,55,62,65,66]. The use of serum galactomannan (GM) antigen was reported in eight cases ([36,47,51,53,57,60,64] and Present Report) and was positive in seven (87.5%) with a median OD index of 2.23; urinary or serum *Histoplasma* antigen was not sought in any of the cases. The most frequently identified species was *Histoplasma capsulatum* (79/113 cases, 69.9%); 17.7% of the cases were attributed to *H. duboisii*.

A majority of patients were treated with one of the amphotericin B formulations followed by itraconazole or other azoles (47/76, 61.8%). Of the 78 patients whose outcomes were recorded, 19 (24.3%) died.

### 3.5. Patients with Other Immunocompromising Conditions (OIC)

There were 24 case reports and 4 case series of 28 patients with OIC and a diagnosis of histoplasmosis [20,44,70,71,72,73,74,75,76,77,78,79,80,81,82,83,84,85,86,87,88,89,90,91,92,93,94]. The patients’ median age was 59 years (range 6–86), and the latency between the last time in an endemic area and diagnosis ranged from 3 weeks to 42 years. The most frequent underlying clinical conditions were solid organ transplantation and autoimmune diseases (51.9%) (Table 4).

All but two of the cases of histoplasmosis (one in Italy and one in Spain) were imported, with 60% imported from Central/South America. The countries reporting the most cases were France (8, 28.6%), Spain (6, 21.4%), and The Netherlands (5, 17.8%). Sixteen patients had progressive disseminated histoplasmosis (57.1%), which was revealed by a picture of haemophagocytic lymphohistiocytosis (HLH) in five cases [76,83,84,88,92]; two had intestinal histoplasmosis [89,91], one of whom was initially misdiagnosed as having Crohn’s disease leading to HLH following treatment with infliximab [91]; the remaining ten had apparently localised disease with cerebral involvement [72], endocarditis [77], pyomyositis [63], or pulmonary [87], oral [78], laryngeal [80], liver [44], osteoarticular [89] or skin diseases [39]. Seven cases (25%) were characterised by ulcerated lips or the involvement of the oropharyngeal mucosa [70,71,72,74,81,84,88].

Cutaneous localisations of *Histoplasma* spp. were reported in nine patients (32.1%) [63,70,73,75,81,83,85,93], presenting as cellulitis [70,92], a solitary nodule [73], or widespread skin lesions [40,63,75,81,82,84]. A majority of cases were diagnosed by means of histology (22, 78.6%) associated with cultures (13, 46.4%) or PCR (10, 35.7%); in two patients, the diagnosis was confirmed by a post mortem examination [20,89]. The use of PCR was reported in 11 cases [44,71,76,84,86,88,90,91,92,93,94]. Two cases were positive for urinary *Histoplasma* antigen [74,75]. GM antigen tests were used in five cases (three in serum, one in bronchoalveolar lavage fluid, and one in bronchoaspirate) affecting four patients [77,79,82,84]; the results were positive in all the samples except for one serum sample [82]

Table 5 shows the main blood alterations observed in the PLWH and the patients with OIC. All the cases were attributed to *H. capsulatum* except for two cases of *H. duboisii* infection [63,82] and one for which the species was not identified [80]. A majority of patients were treated with amphotericin B (16/26, 61.5%), followed by maintenance itraconazole treatment (8/26, 30.8%); eight patients received itraconazole, in one of whom it was associated with nephrectomy. The outcomes of 25 patients were available and included eight deaths (32%).

### 3.6. Immunocompetent Patients

A total of 81 immunocompetent subjects with a diagnosis of histoplasmosis were described in 36 case reports and 13 case series (5 of which also included PLWH) [23,33,38,40,95,96,97,98,99,100,101,102,103,104,105,106,107,108,109,110,111,112,113,114,115,116,117,118,119,120,121,122,123,124,125,126,127,128,129,130,131,132,133,134,135,136,137]. Their median age was 43 years (range 17–78), and there was a prevalence of males (57/79, 72.1%) (Table 6). Median latency was 14 days in the case of travellers (range 2–120) and 6.5 years (range 3–46) in the case of migrants or expatriates. Six cases were considered to be autochthonous (three in Italy and one each in Spain, Ireland, and Israel) [105,106,117,121,126,131]; the Spanish case involved a laboratory technician who accidentally inoculated himself while handling a sample containing *H. capsulatum* [106]. The geographical areas in which the imported cases were acquired were mainly Central and South America (64.5%), followed by Africa (26.3%). Most cases (46.3%) were diagnosed in Spain, France, and Germany. There were six clusters of acute histoplasmosis involving 3–10 travellers to Latin America who had visited bat-infested caves [98,99,100,108,109] and the members of a multinational student expedition to a Ugandan rainforest in which a hollow bat-infested tree was the possible source of infection [115]. Most of the immunocompetent patients had pulmonary histoplasmosis (60/83, 73.2%), which was associated with rheumatological manifestations (arthralgia and/or erythema nodosum) in 23.3% of cases (Table 6).

There were eight cases (13.3%) of single or multiple nodular lung lesions mimicking neoplasia or severe sarcoidosis [95,97,107,119,120,121,128,133], three cases of chronic cavitary histoplasmosis [101,102,127], and one patient with fibrosing mediastinitis [130]. Five patients (6.2%) presented with a picture of PDH [103,116,123,125,126], and two showed central nervous system involvement [103,134]. Oropharyngeal ulcerations (tongue, uvula, gingiva, tonsils, hard palate) were described in five patients, and three had primary cutaneous histoplasmosis. Most of the cases were diagnosed by means of serology (44.4%) followed by histology (12.3%); one patient was diagnosed post mortem [126] (Table 6). The most frequently used drug was itraconazole (60.8% of cases); 25.7% of the patients did not require any treatment. The outcome was apparently favourable in 94.4% of cases.

### 3.7. Histoplasma Capsulatum from Animals in Europe: The One Health Concept

There were seven cases of veterinary histoplasmosis: one in a dorcas gazelle (*Gazella dorcas neglecta*) living in captivity in Spain [137]; three in domestic cats in Italy (two cases) and Austria [138,139,140], one in a European hedgehog (*Erinaceus europaeus*) in Germany [141], one in a badger (*Meles meles*) in Germany [142], and one in a dog in southern Italy [143]. A necropsy study of free-ranging mustelids in Switzerland relating to 566 reports written between 1958 and 2015 documented 6 cases of histoplasmosis in 249 badgers (*Meles meles*) [144]. Except for the gazelle and the cat in Austria (imported from Texas), all the cases can be considered autochthonous.

The disease was disseminated in four cases (the gazelle, two cats, and hedgehog) [137,138,140,141], limited to the skin in one cat [139] and the badger in Germany [142] and involving the spinal cord in the dog [143].

## 4. Discussion

This systematic review of diagnoses of histoplasmosis made in Europe and Israel over 16 years identified 728 cases. Of those, 505 were described in five retrospective studies, including three studies of acute histoplasmosis in travellers diagnosed in Spain [13], Italy [15], and Israel [16]; one study of imported AIDS-related histoplasmosis conducted by the French National Reference Centre for Mycoses and Antifungals (NRCMA) [14]; and one summarising imported endemic mycoses (including histoplasmosis) in hospitalised patients in Spain [12]. The remaining 223 cases were retrieved from single case reports or small case series [17,18,19,20,21,22,23,24,25,26,27,28,29,30,31,32,33,34,35,36,37,38,39,40,41,42,43,44,45,46,47,48,49,50,51,52,53,54,55,56,57,58,59,60,61,62,63,64,65,66,67,68,69,70,71,72,73,74,75,76,77,78,79,80,81,82,83,84,85,86,87,88,89,90,91,92,93,94,95,96,97,98,99,100,101,102,103,104,105,106,107,108,109,110,111,112,113,114,115,116,117,118,119,120,121,122,123,124,125,126,127,128,129,130,131,132,133,134,135,136].

The first important message arising from the review is that only seven cases diagnosed in Europe (1%) [23,70,79,105,106,121,126] and one diagnosed in Israel [131] were not associated with any disclosed history of travel or residence in areas considered to be endemic for the disease and can, therefore, possibly be classified as autochthonous: four in Italy [23,79,105,126], two in Spain [70,106], and one in Ireland [121]. Italy has long been considered a country with endemic foci of histoplasmosis, and a number of autochthonous cases have been previously reported [7,8,145,146,147,148]. Moreover, we retrieved two autochthonous veterinary cases involving a cat and a dog living in Italy, thus confirming that Italy should be considered a low-level endemic region [138,143]. One of the two Spanish cases involved a kidney transplant recipient, and although donor-derived infection cannot be definitely excluded, both the donor and the recipient were born in Spain and had no history of travel to endemic areas [70]; the other involved a laboratory technician who accidentally inoculated himself with a biological sample taken from an Ecuadorian man with a diagnosis of histoplasmosis [106]. A third autochthonous case (described before the beginning of our study period) occurred in an immunocompromised patient treated with azathioprine in Andalusia [149]. We are not aware of any previous autochthonous cases of histoplasmosis diagnosed in Ireland, and the Israeli report (which was the first to be published in the literature) involved a patient who lived near the cave of Yodfat where *H. capsulatum* was isolated from bats [131,150].

Most of the imported cases were acquired in Central and South America (120/219, 54.8%), with Ecuador being the most represented country of acquisition among PLWH (16/50, 32%) and immunocompetent subjects (12/50, 24%). Ghana and Ivory Coast (16/47, 34%) are the two African countries contributing the most patients with histoplasmosis associated with HIV/AIDS, although the NRCMA study of patients diagnosed with histoplasmosis in metropolitan France found that 39% of the cases had been acquired in French Guiana and French West Indies [14]. Among immunocompetent travellers acquiring histoplasmosis in Africa, the most represented country is Uganda, but this is due to a cluster of cases following a study field trip to a rainforest [115].

Of the cases of histoplasmosis diagnosed in Africa, 28 were due to *H. capsulatum* var. *duboisii*, 35% of which were acquired in the Democratic Republic of Congo (DRC) and Congo Brazzaville, which confirms recent review data indicating a 44% cumulative prevalence of histoplasmosis in both countries [10] as well as the findings of a recent study showing that *Hcd* is mainly responsible for histoplasmosis in the DRC [151]. Interestingly, Ghana contributed 15% of the cases included in our review but only one in the review cited above [10].

The fact that Spain (59.9%) and France (20.1%) together account for 80% of the European diagnoses of histoplasmosis is due to large-scale immigration from Latin America to Spain, from French overseas departments (i.e., French Guiana and French West Indies) to France, and from former colonial territories to both [12,13,14]. It is also interesting to note that more than half of the cases diagnosed in The Netherlands have links to Suriname and the Dutch Antilles.

We confirmed that the latency period from the last time spent in an endemic area to the time of diagnosis can be extremely long, except in the case of travellers with acute pulmonary histoplasmosis: median latency is 66 months among PLWH, 6 years among immunocompetent expatriates/migrants, and between as short as 3 weeks [83] and as long as 42 years [78] among patients with OIC. As previously shown by epidemiological studies of endemic areas [152] and among travellers [8], a majority of our cases involved males (440/659 66.8%) [12,14,15,16,17,18,19,20,21,22,23,24,25,26,27,28,29,30,31,32,33,34,35,36,37,38,39,40,41,42,43,44,45,46,47,48,49,50,51,52,53,54,55,56,57,58,59,60,61,62,63,64,65,66,67,68,69,70,71,72,73,74,75,76,77,78,79,80,81,82,83,84,85,86,87,88,89,90,91,92,93,94,95,96,97,98,99,100,101,102,103,104,105,106,107,108,109,110,111,112,113,114,115,116,117,118,119,120,121,122,123,124,125,126,127,128,129,130,131,132,133,134,135,136], although the male-to-female ratio was lower (2:1) than that observed in other studies (4:1) [152].

Underlying diseases or treatments responsible for immunosuppression were present in 59.7% of the subjects, with HIV infection being the main risk factor (83.9%), followed by solid organ transplantation (3.9%). Primary immunodeficiency was diagnosed in 1.4% of the patients [12,20,40,73,86], and was notable for two cases of idiopathic CD4+ lymphocytopenia [40,73] and one case of IFN-γ autoantibodies identified during the patient’s follow-up [71,86]. In the case of travellers, the most frequent route of exposure to *Histoplasma* was the exploration of bat-ridden caves and/or contact with bat guano, which was described in 52.2% of the cases for which this information was available [15,33,99,100,108,109,112,114,115,124,133,134,136], thus confirming this well-known risk factor [6,8], which should always be investigated in subjects presenting with respiratory symptoms, especially if they have recently travelled to Latin America [153].

Progressive disseminated histoplasmosis (PDH) was the rule among the PLWH (90.3%) and was also observed in more than a half of the patients with OIC (57.1%) and 6.2% of the immunocompetent patients. This last prevalence is in line with the 8% found in a recent systematic review of acute histoplasmosis in immunocompetent travellers [9]. One recent retrospective study of 261 cases conducted in the United States found similar prevalence of disseminated disease among PLWH (78%) and patients with OIC (58%), but a higher prevalence among immunocompetent subjects (33%) [154]. As the exposure of the majority of the immunocompetent patients considered in our review was limited to travel, it can be speculated that the higher rate of disseminated disease in the American review may have been due to greater inoculum exposure.

A diagnosis of histoplasmosis was the AIDS-revealing condition in 52% of our PLWH and was associated with severe immunodepression as confirmed by the very low median CD4+ cell count and the observation of concomitant opportunistic infections in nearly 30% of the patients. Although gastrointestinal involvement was less frequent than in highly endemic areas such as French Guiana [155,156], it is worth noting that 5.3% of the patients had localised intestinal histoplasmosis, and that the fungus was visualised and/or cultured in 10.5% of the cases. Skin lesions were described in 32% of PLWH, 21.4% of the patients with OIC, and 4.9% of the immunocompetent patients; furthermore, primary cutaneous histoplasmosis was observed in an HIV-positive patient and an immunocompetent laboratory technician [39,106]. Oral ulcers in isolation or as part of disseminated disease [132] were relatively frequent among the PLWH (5.3%) and the patients with OIC (25%). Morote et al. found mucocutaneous involvement in 8.8% of their PLWH in French Guiana, which suggests an association with more profound immunosuppression and a risk of early death [157]. Interestingly, haemophagocytic lymphohistiocytosis (HLH) was described more frequently among the patients with OIC (14.2%) than the PLWH (2.6%), which seems to be in line with the finding of only 34 cases of secondary HLH associated with HIV infection in Cayenne Hospital [158].

Histology and cultures were the most frequently used means of diagnosing histoplasmosis in the immunocompromised PLWH and patients with OIC, whereas antibody detection was more frequently used in the diagnostic work-up of immunocompetent patients, thus confirming the data of Staffolani et al. [9]. Although a search for serum and urinary *Histoplasma* antigen is considered a very sensitive and rapid means of diagnosing histoplasmosis, it was used in none of the PLWH and only two of the patients with OIC [73,74], but in five immunocompetent patients (6.1%) [103,124,136]. In order to overcome the very limited availability of the *Histoplasma* antigen in most European laboratories, galactomannan (GM) antigen is used as a surrogate marker of infection on the grounds of its cross-reactivity with *Histoplasma capsulatum* antigen [159] and was positive in 87.5% of the PLWH ([[36],[47],[51],[53],[57][60],[64]] and PR) and 75% of the patients with OIC [77,79,82,84] who had disseminated histoplasmosis. The use of PCR was reported in 33.2% of the patients as a whole: 46/114 PLWH (40.3%) (Table 2) and 11/28 patients with OIC (39.3%) (Table 4). It was mainly used to complement the traditional methods of microscopy and cultures, and the diagnosis of only three PLWH was exclusively based on PCR [38]. As no details are given concerning PCR protocols and gene targets used, it is difficult to draw any definite conclusions about the role of PCR in the diagnosis of histoplasmosis, especially in non-endemic countries [160]; however, it is worth noting that a recent French comparative study has found that using real-time quantitative PCR and the ribosomal small subunit RNA (*mtSSU*) gene of *H. capsulatum* as a target has the advantage of a high sensitivity (97.7%) [161].

Most of the immunocompromised patients with PDH were treated with amphotericin B followed by itraconazole in accordance with the guidelines but, as shown in Table 2 and Table 4, a number of other regimens were used, even though they are usually considered less effective [162]. Sixty percent of the immunocompetent subjects with acute pulmonary histoplasmosis were apparently successfully treated with itraconazole. It is difficult to evaluate clinical outcomes on the basis of single case reports and small case series, but mortality rates in the different categories of patients (PLWH 24.1%; patients with OIC 32%; and immunocompetent patients 5.6%) were sufficiently similar to those reported by Franklin et al. in the USA (15%; 24%, and 13%, respectively) [154].

This systematic review has two limitations: because of the exclusion of some languages (Turkish, Finnish, Romanian, and Russian), it is possible that some published cases were not considered, and the absence of some important information from the case reports and small case series may have biased our findings.

## 5. Conclusions

In conclusion, this review highlights the fact that histoplasmosis should be considered in the differential diagnosis of systemic diseases among immunocompromised subjects in Europe. The potentially long latency period in such patients requires a more extensive evaluation of their travelling history than that required in immunocompetent travellers presenting with respiratory symptoms.

## Figures and Tables

**Figure 1 jof-07-00481-f001:**
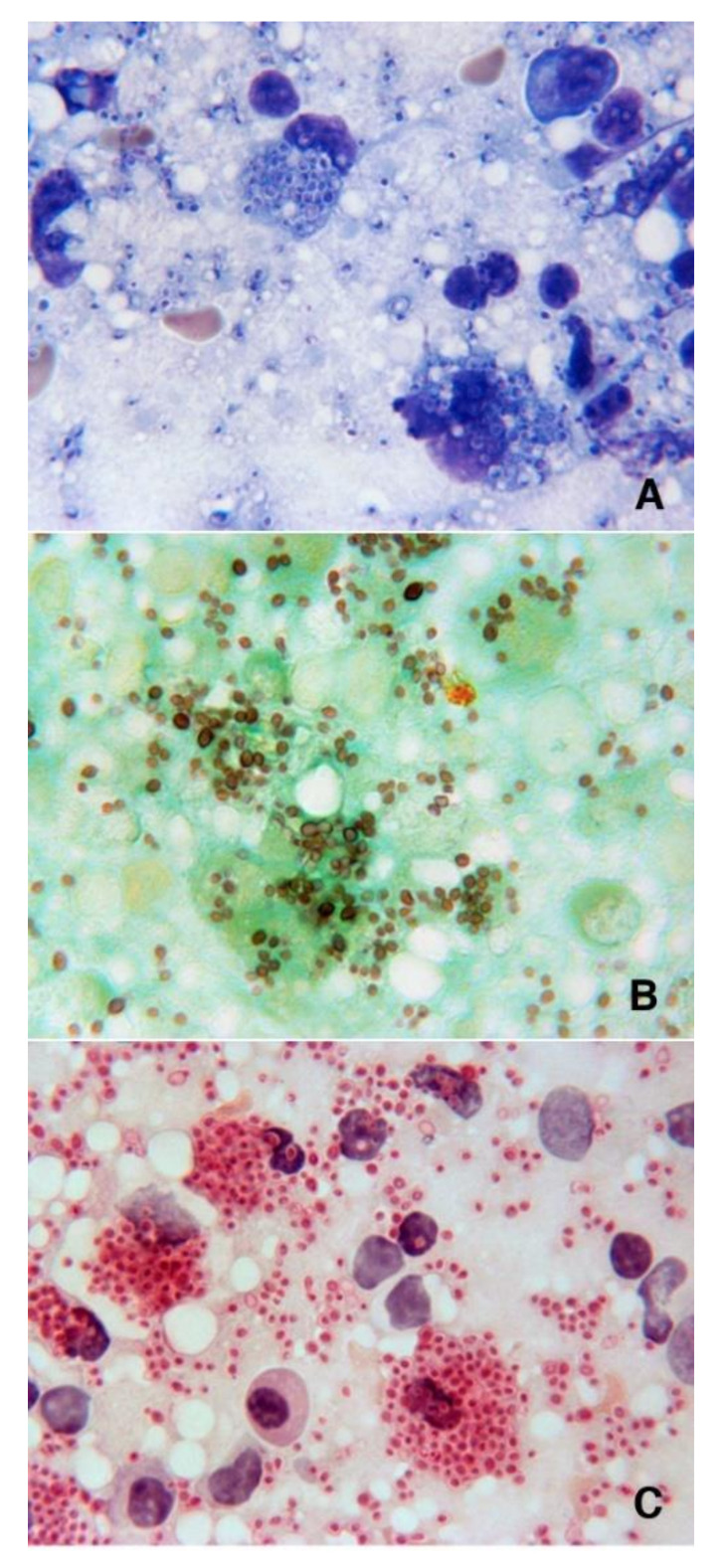
Imprint touch preparations from latero-cervical lymph node biopsies stained with Giemsa (**A**), Grocott (**B**), and PAS (**C**), showing intracytoplasmic capsulated microorganisms morphologically consistent with *Histoplasma* spp. (magnification 1000×).

**Figure 2 jof-07-00481-f002:**
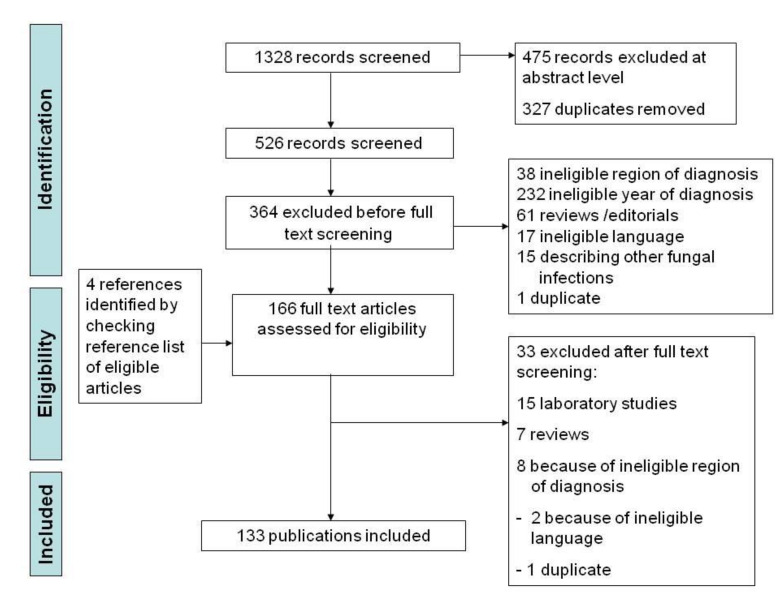
Modified PRISMA flow chart.

**Table 1 jof-07-00481-t001:** Studies describing 505 cases of histoplasmosis in Europe and Israel between 2005 and 2020.

Author, Year, Reference	Type of Study	Country	Period of Study	No. of Patients Diagnosed	Age Median Years (Range)	Male Gender	Risk Factors	Disease	Identified *Histoplasma* Species	Outcomes
Gascon, 2005, [13]	Prospective, single centre	Spain	Mar. 2001– Apr. 2003	69/342 (20.2%) 13/69 (19%) symptomatic	NR	NR	Travellers to Central/South America	Acute pulmonary disease	NR	NR
Molina-Morants, 2018, [12]	Retrospective, patients admitted to Spanish hospitals	Spain	Jan. 1997– Dec. 2014	286 (incidence: 0.53/100,000)	37 (30–50)	193 (67.5%)	188 immunodeficiency (65.7%) :151 HIV (80.7%); 14 solid neoplasia (7.5%); 10 solid organ transplant (5.3%); 9 haematological malignancy (4.8%);5 systemic autoimmune disease (2.7%); 5 end-stage renal disease (2.7%); 3 cirrhosis (1.6%); 2 primary immune deficiency (1.1%)	72 isolated pulmonary disease (25.2%); 36 extra- pulmonary disease (12.6%); 168 unspecified disease (58.7%)	10 *H. duboisii*	44 deaths (15.4%)
Peigne, 2011, [14]	Retrospective, NRCMA	France	1985– 2006	104	38 * 40 **	31 (78%) * 37 (58%) **	HIV/AIDS	73 PDH (70%); blood (36.5%); bone marrow (61.5%); skin (38.5%); lymph nodes (24%); respiratory samples (30.8%); CNS (1.9%)	NR	7 IRIS (11%)41 deaths (39.4%) (median follow-up: 31.5 months)
Segel, 2015, [16]	Retrospective, single centre	Israel	2000– 2011	23	31	17 (73.9%)	Travellers to Central/South America (95.6%); 14 exposure to bat habitats (61%)	Symptomatic pulmonary disease 60.9%	2 *H. capsulatum*	NR
Staffolani, 2020, [15]	Retrospective study, single centre	Italy	Jan. 2005– Dec. 2015	23	38.5 Ecuador cluster; 46.7 others	17 (73.9%)	17 Scientific expedition (Ecuador): bat excreta; 3 speleologist; 2 tourism, 1 work (Panama, Bolivia, Mexico, Cuba)	2 PDH (1 immunocompetent, 1 immunocompromised) 21 pulmonary disease	2 *H. capsulatum*	All alive

HIV—human immunodeficiency virus; PDH—progressive disseminated histoplasmosis; NRCMA—French National Reference Centre for Mycoses and Antifungals; IRIS—immune reconstitution syndrome; CNS—central nervous system; NR—not reported; * pre-highly active antiretroviral therapy (HAART) era; ** HAART era.

**Table 2 jof-07-00481-t002:** Characteristics of 114 patients * living with HIV/AIDS diagnosed as having histoplasmosis in Europe.

Characteristic	No. (%)
Median age, years (range)	37 (2–63)
Males	73 (64)
Previously known HIV infection	37 (32.4)
Histoplasmosis indicating HIV infection	40 (35.1)
Time of HIV diagnosis unknown	37 (32.4)
Median latency, months (range)	66 (1–420)
Median CD4+ cells/μL	17 (0–594)
**Area of suspected *Histoplasma* exposure**	
Central/South America ^§^	55 (50)
Sub-Saharan Africa ^^^	47 (42.7)
Asia ^#^	6 (5.5)
United States/Mexico	1 (0.9)
Europe	1 (0.9)
Not reported	4 (3.5)
**Country of diagnosis**	
Spain	56 (49.1)
France	22 (19.3)
Italy	14 (12.3)
Switzerland, The Netherlands	6 (5.2)
United Kingdom	4 (3.5)
Portugal	2 (1.7)
Belgium, Denmark, Finland, Germany	1 (0.9)
***Histoplasma*** **species**	
*H. capsulatum*	80 (70.2)
*H. duboisii*	20 (17.5)
*Histoplasma* spp.	14 (12.3)
**Concomitant infections/neoplasia**	**33 (28.9)**
Fungal	
Oral candidiasis	8 (24.4)
Oesophageal candidiasis	2 (6.1)
PJP	5 (15.1)
Cryptococcosis	2 (6.1)
Coccidioidomycosis	1 (3.0)
Bacterial	
Tuberculosis	2 (6.1)
MAC	3 (9.1)
*Salmonella bacteremia*	1 (3.0)
Protozoal	
Chagas disease	1 (3.0)
Isopsoriasis	1 (3.0)
Helminthic	
*Strongyloides stercoralis*	3 (9.1)
Neoplasia	
Kaposi’s sarcoma	3 (9.1)
Multiple	8 (7.0)
**Methods of diagnosis**	
Histology + culture	32 (28.1)
Culture + PCR	19 (16.7)
Histology	16 (14.0)
Culture	14 (12.3)
Histology + PCR	13 (11.4)
Histology + culture + PCR	7 (6.1)
Histology + culture + blood smear	5 (4.4)
Histology + PCR	2 (1.7)
PCR	3 (2.6)
Histology + PCR + blood smear	1 (0.9)
Autopsy	2 (1.7)
**Treatment**	
L-AMB, itraconazole	30 (38)
d-AMB, itraconazole	12 (15.2)
Itraconazole	14 (12.7)
L-AMB	8 (10.1)
L-AMB, voriconazole or posaconazole	3 (3.8)
ABLC, itraconazole, fluconazole	3 (3.8)
Fluconazole	3 (3.8)
ABLC	2 (2.5)
d-AMB	2 (2.5)
Not treated	2 (2.5)
Not reported	35
**Outcome**	
Survived	60 (75.9)
Died	19 (24.1)
Not reported	35

* Including our case report. PJP: *Pneumocystis jirovecii* pneumonia; MAC: *Mycobacterium avium-intracellulare*; PCR—polymerase chain reaction; L-AMB—liposomal amphotericin B; d-AMB—deoxycholate amphotericin B; ABLC—amphotericin B lipid complex. ^§^ 16 Ecuador; 7 Colombia; 5 Brazil; 3 French Guiana, Bolivia, Peru, Suriname; 2 Venezuela, Paraguay; 1 Cuba, Haiti, Martinique, Trinidad and Tobago, Nicaragua, Panama, Dutch Antilles. ^^^ 9 Ghana; 7 Ivory Coast; 5 Cameroon; 4 Senegal, Nigeria; 3 Democratic Republic of Congo, Congo; 2 Equatorial Guinea, Liberia; 1 Togo, Tanzania, Guinea–Conakry. ^#^ 2 Cambodia, Thailand; 1 Malaysia.

**Table 3 jof-07-00481-t003:** Summary of the signs and symptoms of histoplasmosis in people living with HIV or other immunocompromising conditions, and the organs in which *Histoplasma* spp. was demonstrated.

People Living with HIV		Other Immunocompromising Conditions	
Signs and Symptoms, No. (%)	Organs in which *Histoplasma* was Demonstrated, No.	Signs and Symptoms, No. (%)	Organs in which *Histoplasma* was Demonstrated, No.
Fever, 57/78 (73.1) Splenomegaly, 22/32 (65.6) Hepatomegaly, 21/32 (65.6) Lymph node enlargement, 40/78 (51.3) Weight loss, 38/78 (48.7) Skin lesions, 25/78 (32.0) Cough, 20/78 (25.6) Dyspnoea, 17/78 (21.8) Diarrhoea, 14/78 (17.9) Abdominal pain, 10/78 (12.8) Nausea, 5/78 (6.4)	Bone marrow, 38 Blood, 29 Lymph nodes, 29 Lung, 24 Skin, 20 Intestine, 12 Liver, 8 Peripheral blood smear, 5 Oropharyngeal mucosa, 5 Cerebrospinal fluid, 2 Larynges, 2 Tonsils, 2 Pleural fluid, 1 Vagina, 1 Brain biopsy, 1 Oesophagus, 1	Fever, 17/26 (65.4) Weight loss, 11/26 (42.3) Cough, 9/26 (34.6) Dyspnoea, 7/26 (26.9) Diarrhoea, 6/26 (23.1) Skin lesions, 5/26 (19.2) Hepatomegaly, 4/26 (15.4)Lymph node enlargement, 4/26 (15.4) Abdominal pain, 3/26 (11.5) Splenomegaly, 1/26 (3.8) Nausea, 0/26 (0)	Lung, 8 Bone marrow, 6 Oropharyngeal mucosa, 6 Lymph nodes, 6 Skin, 5 Intestine, 4 Liver, 3 Kidney, 2 Brain biopsy, 1 Muscle, 1 Spleen, 1 Synovial membrane, 1 Blood, 1 Mitral valve, 1

**Table 4 jof-07-00481-t004:** Characteristics of 28 immunocompromised HIV-negative patients diagnosed as having histoplasmosis in Europe.

Characteristics	No. (%)
Median age, years (range)	59 (6–86)
Males	16 (57.1)
Latency interval	3 months–42 years
**Geographical area of suspected *Histoplasma* exposure**	
Central/South America *	14 (50.0)
Sub-Saharan Africa ^§^	5 (12.9)
Asia ^^^	2 (7.1)
Sub-Saharan Africa/Central America ^°^	1 (3.6)
South America/Asia **	1 (3.6)
Asia/Central America ^#^	1 (3.6)
Asia (Malaysia)/United States	1 (3.6)
South America (Suriname)/United States	1 (3.6)
Europe (Italy, Spain)	2 (7.1)
**Country of diagnosis**	
France	8 (28.6)
Spain	6 (21.4)
Netherlands	5 (17.9)
Germany	3 (10.7)
United Kingdom	2 (7.1)
Belgium, Italy, Portugal, Sweden	1 (3.6)
**Underlying diseases/immunosuppressive treatment**	
SOT (kidney 2, kidney/liver 1, liver 2, lung 1, NR 1)	7 (25.0)
Autoimmune diseases (4 RA, 1 SLE, 1 dermatomyositis,	7 (25.0)
1 myasthenia gravis)	
Primary immunodeficiency (IgA deficit 1, idiopathic CD4 lymphopenia 2, autoantibodies against IFN-γ 1)	4 (14.3)
Sarcoidosis	
Chronic lymphocytic leukaemia	2 (7.1)
Cancer	2 (7.1)
Ulcerative colitis	2 (7.1)
Haemodialysis	1 (3.6)
Steroids	1 (3.6)
	2 (7.1)
**Methods of diagnosis**	
Histology + culture	8 (28.6)
Histology + PCR	5 (17.8)
Histology + culture + PCR	5 (17.8)
Histology	4 (14.3)
Culture	3 (10.7)
Culture + PCR	1 (3.6)
Autopsy	2 (7.1)
***Histoplasma*** **species**	
*H. capsulatum*	25 (89.3)
*H. duboisii*	2 (7.1)
*Histoplasma* spp.	1 (3.6)
**Treatment**	
d-AMB, itraconazole	5 (19.2)
L-AMB, itraconazole	5 (19.2)
Itraconazole	5 (19.2)
L-AMB	3 (11.5)
ABLC, itraconazole	1 (3.8)
L-AMB, posaconazole	1 (3.8)
Voriconazole, itraconazole	1 (3.8)
Fluconazole, micafungin, L-AMB	1 (3.8)
Fluconazole	1 (3.8)
Not treated	1
Not reported	1
**Outcome**	
Survived	17 (68.0)
Died	8 (32.0)
Not reported	3

* 3 Ecuador, Suriname; 2 Brazil; 1 French Guiana, Mexico, Venezuela, Nicaragua, Costa Rica; 2 country not reported. ^§^ 1 Guinea Bissau; 2 country not reported; 2 multiple countries. ^°^ Cuba, Cameroon, Chad. ** Suriname, Thailand, Malaysia. ^#^ Thailand, Costa Rica. ^^^ 1 Thailand, Bangladesh. SOT—solid organ transplantation; NR—not reported; RA—rheumatoid arthritis; SLE—systemic lupus erythematosus; d-AMB—deoxycholate amphotericin B; L-AMB—liposomal amphotericin B; ABLC—amphotericin B lipid complex.

**Table 5 jof-07-00481-t005:** The number of patients living with HIV (PLWH) or other immunocompromising conditions (OIC) undergoing the main blood tests.

Blood Test	PLWH	OIC
Haemoglobin, No.	38	12
Anaemia (<12 g/dL), No. (%)	36 (94.7)	12 (100)
Median Haemoglobin value, g/dL (range)	7.8 (3.9–12.3)	9.8 (5.3–11.6)
White blood cells, No.	37	14
Leukopenia (<4000/μL), No. (%)	26 (70.3)	6 (42.8)
Median white blood cells value, cells/L (range)	3800 (960–13,600)	5545 (1400–16,760)
Platelets, No.	27	8
Thrombocytopenia (<150,000/μL)	19 (70.4)	3 (37.5)
Median platelets value/μL	80,000 (20,000–272,000)	170,500 (17,000–579,000)
AST, No.	16	7
Median AST value, UL (range)	97.5 (50–610)	108 (9–1046)
ALT, No.	14	9
Median ALT value, UL (range)	63 (27–301)	58 (8–487)

PLWH—people living with HIV; AST—aspartate aminotransferase; ALT—alanine aminotransferase.

**Table 6 jof-07-00481-t006:** Characteristics of 81 cases of histoplasmosis among immunocompetent subjects diagnosed in Europe and Israel.

Characteristics	No. (%) or Median (Range)
Age, years	43 (17–78)
Males	56 (69.1)
Latency	
Travellers	14 days (2–120)
Expatriates, migrants	6.5 years (3–46)
**Geographical areas of suspected *Histoplasma* exposure**	
Central/South America ^^^	50 (61.7)
Sub-Saharan Africa ^§^	20 (24.7)
Sub-Saharan Africa and Central/South America ^#^	3 (3.7)
United States *	2 (2.5)
India	1 (1.2)
Europe (2 Italy, 2 Spain)	4 (4.9)
Israel	1 (1.2)
**Country of diagnosis**	
Spain	19 (23.5)
France	12 (14.8)
Germany	8 (9.9)
Italy, Poland	6 (7.4)
Austria, Slovenia	5 (6.2)
Netherlands	4 (4.9)
Israel, United Kingdom, Switzerland	3 (3.7)
Greece, Ireland, Portugal	2 (2.5)
Sweden	1 (1.2)
**Patient categories**	
Travellers	40 (49.4)
Expatriates	5 (12.3)
Workers	26 (32.1)
Migrants	2 (2.5)
Military service people	2 (2.5)
Autochthonous	5 (7.4)
**Method of diagnosis**	
Serology	36 (44.4)
Histology	10 (12.3)
Histology + PCR	9 (11.1)
Histoplasma antigen	4 (4.9)
Histology + culture + PCR	3 (3.7)
Culture	3 (3.7)
Culture + PCR	3 (3.7)
Histology + culture	2 (2.5)
PCR	2 (2.5)
Serology + PCR	2 (2.5)
Clinical	6 (7.4)
Autopsy	1 (1.2)
***Histoplasma*** **species**	
*H. capsulatum*	29 (78.4)
*H. duboisii*	6 (16.2)
*Histoplasma* spp.	2 (2.5)
**Treatment**	
Itraconazole	45 (60.8)
L-AMB	6 (8.1)
Ketoconazole, itraconazole	2 (2.7)
d-AMB, itraconazole	2 (2.7)
No treatment	19 (25.7)
Not reported	7
**Outcome**	
Survived	68 (94.4)
Died	4 (5.6)
Not reported	9

^^^ 12 Ecuador; 6 Venezuela; 4 Nicaragua, Brazil, Mexico; 3 Costa Rica, Cuba, Trinidad; 2 Jamaica; 1 Guadeloupe, Guatemala, Peru, Panama, El Salvador; 6 multiple countries. ^#^ 1 Brazil, Bolivia, Angola, Ethiopia. ^§^ 12 Uganda; 2 Democratic Republic of Congo, Guinea Bissau; 1 Angola, Congo, Ghana, Equatorial Guinea, Gabon. * 1 also Mexico. PCR—polymerase chain reaction; L-AMB—liposomal amphotericin B; d-AMB—deoxycholate amphotericin B.

## Data Availability

No new data were created or analysed in this study. Data sharing does not apply to this article.

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
