# Peer review of "Histoplasmosis Diagnosed in Europe and Israel: A Case Report and Systematic Review of the Literature from 2005 to 2020"

_jof, 2021, doi:10.3390/jof7060481_

Round 1
Reviewer 1 Report
The review contains interesting tables and the discussion is well written.
I have few comments:
- The species names are written either as H. capsulatum var. capsulatum or as H. capsulatum and H. capsulatum var. duboisii and H. duboisii. Explain the nomenclature of this fungus.
- Use the formal name ‘HHV-8’ instead of ‘KSHV’ viremia
- Use ‘.’ instead of ‘,’ for decimal numbers
- Table 2. Reformulate the line ‘not reported’ 37 (32.4%).
- For the molecular diagnosis, discuss the different approaches i.e. a specific Histoplasma PCR versus panfungal PCR followed by sequencing.
- Table 3. Rephrase the title.
- Table 4. Make clear in the title that it concerns non-HIV patients
- Table 5. Blood parameters are only shown for a subset of patients. Was this information not available for the other patients? Are these the blood parameters on admission of the patients?
Author Response
Reviewer # 1
The review contains interesting tables and the discussion is well written.
I have few comments:
- The species names are written either as H. capsulatum var. capsulatum or as H. capsulatum and H. capsulatum var. duboisii and H. duboisii. Explain the nomenclature of this fungus
Thanks for the comment; we have corrected the nomenclature (see also response to reviewer 2)
.
- Use the formal name ‘HHV-8’ instead of ‘KSHV’ viremia. Done
- Use ‘.’ instead of ‘,’ for decimal numbers
Thanks for this comment; We have changed along all the manuscript.
- Table 2. Reformulate the line ‘not reported’ 37 (32.4%). We have reformulated the line as “Time of HIV diagnosis unknown”
- For the molecular diagnosis, discuss the different approaches i.e. a specific Histoplasma PCR versus panfungal PCR followed by sequencing. We choose not to discuss this issue because the majority of case reports either use home-made PCR or the authors do not detail the PCR used. We believe that the sentence in our discussion (line 448 page 16) is explicative
- Table 3. Rephrase the title. Done
- Table 4. Make clear in the title that it concerns non-HIV patients OK (we have inserted HIV-negative)
- Table 5. Blood parameters are only shown for a subset of patients. Was this information not available for the other patients? Yes blood parameteres were available only for a minority of patients. Are these the blood parameters on admission of the patients? The answer is yes
Reviewer 2 Report
General comments
This work by Antinori et al., is a fine review of all the histoplasmosis cases described in several european countries and in Israel in the last 15 years. One new case report is described and the literature cases are divided into three different categories: Histoplasmosis in people living with HIV, other immunocompromised patients, and immunocompetent patients. Some veterinary cases are also discussed.
I have noticed throughout the text a confusion in the nomenclature in the agents causing histoplasmosis. It is not clear, when reading the manuscript, if Histoplasma capsulatum has two variants i.e. capsulatum and duboisii, or if that they are two different species (H. capsulatum and H. duboisii). Anyway, the taxonomy of the genus Histoplasma has changed in the recent years and more species have been described and reorganized. Although I understand that H. capsulatum and H. duboisii are nomenclatures more common in the clinical arena, the correct names should be used. It would be worth making a comment about this in the introduction.
Punctuation marks, especially commas, are lacking throughout the manuscript. Please, review carefully.
Add a dot, or a dot followed by a comma, after “spp” and “et al”.
Specific comments
Line 131: correct “microorganisms”.
Line 133: Introduce PCR in line 133 and remove “polymerase chain reaction” from lines 228 and 440.
Line 135: replace “patients” by “patient”.
Line 233: in the introduction, it is stated that two variants of H. capsulatum are the causative agents of histoplasmosis. However, in line 234 are shown as different species. Please, clarify (see general comments).
Line 314: What does “the One Health concept” mean?
Line 406: replace “an” by “a” before “higher”.
Line 409: replace “an” by “a” before “high” and remove the “-“ between “high” and “inoculum”.
Line 448: delete “an”
Author Response
Reviewer # 2
This work by Antinori et al., is a fine review of all the histoplasmosis cases described in several European countries and in Israel in the last 15 years.
I have noticed throughout the text a confusion in the nomenclature in the agents causing histoplasmosis. It is not clear, when reading the manuscript, if Histoplasma capsulatum has two variants i.e. capsulatum and duboisii, or if that they are two different species. Anyway the taxonomy of the genus Histoplasma has changed in the recent years and more species have been described and reorganized. Although I understand that H. capsulatum and H. duboisii are nomenclatures more common in the clinical arena, the correct names should be used. It would be worth making a comment about this in the introduction.
We thanks the reviewer for her/his general comment. She/he is correct about the confusion between the term species and varieties referred to H. capsulatum and H. duboisii. We have correct and adopted the term “variety” to indicate both. As correctly pointed out by the reviewer in the clinical arena this is the nomenclature adopted. However, in response to her/his criticism we have added in the introduction two sentences reporting the new proposed taxonomy by Sepulveda et al (mBio 2017, 8, e01339-17) which consider four species from the Americas (H. capsulatum sensu stricto, H. mississippiense sp. nov, H. ohiense sp. nov, H. suramericanum sp. nov) and a possible species from Africa (H. duboisii). Although we believe that taxonomy of Histoplasma should be considered a “work in progress” this added part can be useful for clinicians.
Punctuation marks, especially commas, are lacking throughout the manuscript. Please review carefully.
This is correct!. We usually apply for editing our manuscripts by an English language editor before submitting to a Journal. However in this case due to the 15 May 2021 deadline we choosed to postpone English revision/editing following revision made after reviewer’s suggestions. Now the manuscript has been carefully revised by and English mother-tongue.
Add a dot, or a dot followed by a comma, after “spp” and “et al”. We have done it.
Specific comments:
Line 131 “microorganisms”. Done
Line 133 PCR introduced, line 228 and 440 removed “polymerase chain reaction”
Line 135 replace “patients” by “patients” Done
Line 233 We have clarified
Line 314: What does the “One health concept” mean. One Health concept recognizes that the health of humans is connected to the health of animals and the environment and it involves applying a collaborative approach to address potential or existing risks that originate at environment-animal-human ecosystems interface.
Line 406: replace “an” by “a” before higher. Done
Line 409: replace “an” by “a” before “high” and remove the – between “high” and “inoculum”. Done
Reviewer 3 Report
The manuscript by Antinori and colleagues is literature review of cases of histoplasmosis found in Europe and Israel from 2005 to 2020. The authors have gone through the medical literature to find all papers that have reported cases of this fungal disease in this part of the world. There have been several other reviews as indicated by the authors although none as contemporary as this one. While the review is reasonably done, there is little new information here. The authors point out that most of the cases are imported and that is no surprise. That most of the cases develop in immunocompromised hosts. Again that is not new information. Very few of their cases are indigenous to this region of the world. Thus, while there is little to criticize about the information in the review, there is little that really distinguishes it. For example, the authors state that the first important message is that only a handful of cases were those who had no travel. That is not new information. I think the authors need to really highlight what new information they are providing. At present, I see very little.
Author Response
While the review is reasonably done, there is little new information here. The authors point out that most of the cases are imported and this is no surprise. That most of the cases develop in immunocompromised hosts. Again that is not new information. Very few of their cases are indigenous to this region of the world. Thus while there is little to criticize about the information in the review, there is little that really distinguished.
We agree with the reviewer’s comment. However, we believe that the scope of a review is to summarize for the general readers information that are generally sparse (as in this case9 in a myriad of case reports. Our aim was to update information about histoplasmosis in Europe and Israel and raise awareness about this fungal infection among European physicians. Moreover, we believe that information about immunocompromised HIV-negative subjects have never been reviewed previously in Europe. The same for veterinary cases (that are a possible link for autochthonous cases observed in Europe). Finally, we have provided more information about epidemiological and clinical data regarding H. duboisii for which published data are limited as highlighted by two recently published reviews (Amona FM et al. Plos Negl trop Dis 2021; Develoux M, et al. Clin Infect Dis 2020).
Round 2
Reviewer 3 Report
The authors have responded. I am not sure how much interest there is in this topic. Nevertheless, we will find out.